



# A global compilation of coccolithophore calcification rates

Chris J. Daniels[1], Alex J. Poulton[1,2], William M. Balch[3], Emilio Marañón[4], Tim Adey[5], Bruce C. Bowler[3], Pedro Cermeño[6], Anastasia Charalampopoulou[5], David W. Crawford[7,8], Dave Drapeau[3], Yuanyuan Feng[9], Ana Fernández[4], Emilio Fernández[4], Glaucia M. Fragoso[10], Natalia González[11], Lisa M. Graziano[3], Rachel Heslop[5], Patrick M. Holligan[5], Jason Hopkins[3], María Huete-Ortega[12], David A. Hutchins[13], Phoebe J. Lam[14], Michael S. Lipsen[15], Daffne C. López-Sandoval[16], Socratis Loucaides[1,5], Adrian Marchetti[17], Kyle M.J. Mayers[5], Andrew P. Rees[18], Cristina Sobrino[4], Eithne Tynan[5], Toby Tyrrell[5].

[1] National Oceanography Centre, Southampton, SO14 3ZH, UK

[2] The Lyell Centre for Earth and Marine Sciences and Technology, Heriot-Watt University, Edinburgh, EH14 4AS, UK

[3] Bigelow Laboratory for Ocean Sciences, East Boothbay, ME 04544, Maine, USA

[4] Departamento de Ecología y Biología Animal, Universidad de Vigo, 36310 Vigo, Spain

[5] Ocean and Earth Science, National Oceanography Centre Southampton, University of Southampton, SO14 3ZH, UK

[6] Institut de Ciencies del Mar, CSIC, E-08003 Barcelona, Spain

[7] Climate Chemistry Laboratory, Institute of Ocean Sciences, Fisheries and Oceans Canada, P.O. Box 6000, Sidney, BC, Canada

[8] Department of Biology, University of Victoria, Victoria, British Columbia BC V8P 5C2, Canada

[9] Tianjin University of Science and Technology, Tianjin Shi, 300457, China

[10] Trondhjem biological station, Department of Biology, Norwegian University of Science & Technology, NO-7491 Trondheim, Norway

[11] Biodiversity and Conservation Area, Universidad Rey Juan Carlos, E-28933 Madrid, Spain

[12] Department of Plant Sciences, University of Cambridge, CB2 3EA, UK

[13] Department of Biological Sciences, University of Southern California, Los Angeles, CA 90089, USA

[14] Department of Ocean Sciences, University of California, Santa Cruz, California, CA 95064, USA

[15] University of British Columbia, Department of Botany, Vancouver, British Columbia BC V6T 1Z4, Canada

[16] Red Sea Research Center, King Abdullah University of Science and Technology, Thuwal 23955-6900, Saudi Arabia

[17] Department of Marine Sciences, University of North Carolina at Chapel Hill, North Carolina, NC 27599, USA

[18] Plymouth Marine Laboratory, Prospect Place, West Hoe, Plymouth, PL1 3DH, UK

*Correspondence to*: Alex J. Poulton (a.poulton@hw.ac.uk)

**Abstract.** The biological production of calcium carbonate ($CaCO_3$), a process termed calcification, is a key term in the marine carbon cycle. A major planktonic group responsible for such pelagic $CaCO_3$ production (CP) are the coccolithophores, single-celled haptophytes that inhabit the euphotic zone of the ocean. Satellite-based estimates of areal CP are limited to open-ocean waters, with current algorithms utilising the unique optical properties of the cosmopolitan bloom-forming species *Emiliania huxleyi*, whereas little understanding of the optical properties and environmental responses by species other than *E. huxleyi* are currently available to parameterise algorithms or models. To aid future areal estimations and validate future modelling efforts we have



constructed a database of 2765 CP measurements, the majority of which were measured using 12 to 24 h
incorporation of radioactive carbon ($^{14}$C) into acid-labile inorganic carbon ($CaCO_3$). We present data collated

from over 30 studies covering the period from 1991 to 2015, sampling the Atlantic, Pacific, Indian, Arctic and
Southern oceans. Globally, CP in surface waters (<20 m) ranged from 0.01 to 8398 µmol C m$^{-3}$ d$^{-1}$ (with a
geometric mean of 16.1 µmol C m$^{-3}$ d$^{-1}$). An integral value for the upper euphotic zone (herein surface to the depth
of 1% surface irradiance) ranged from <0.1 to 6 mmol C m$^{-2}$ d$^{-1}$ (geometric mean 1.19 mmol C m$^{-2}$ d$^{-1}$). The full
database is available for download from PANGAEA as doi: 10.1594/PANGAEA.888182.


**Keywords.** Calcification, coccolithophores, $CaCO_3$ production.

## 1   Introduction

The formation, export and burial of $CaCO_3$ is an important component of the oceanic carbon cycle, with the

combination of the first two providing a positive feedback on atmospheric $CO_2$ (Archer, 1996; Sarmiento et al.,
2002; Berelson et al. 2007). Around half of oceanic $CaCO_3$ production occurs in shallow neritic environments,
with the remainder occurring in the upper waters of the open-ocean (Milliman, 1993). Export and deep-sea burial
in the open-ocean are both relatively well characterised in terms of global magnitude (Milliman, 1993; Berelson
et al., 2007) and regional trends (e.g. Archer, 1996; Henson et al., 2012), and are often (simply) parameterised in

global biogeochemical models (e.g. Gehlen et al., 2007; Yool et al., 2013) as a function of carbonate chemistry.
The scale of biological formation of $CaCO_3$ in the upper ocean, however, is poorly constrained, in terms of both
its magnitude and biogeography (Berelson et al., 2007), due to knowledge gaps existing in the ecological and
physiological understanding which is fundamental to allow accurate or reliable parameterisation at a global scale
(Balch et al., 2007; Monteiro et al., 2016; Hopkins and Balch, 2018).


Problems with forming such a global perspective on pelagic $CaCO_3$ production partly arise due to the diversity of
the different planktonic organisms involved (coccolithophores, foraminifera, pteropods, and, to a lesser extent
some dinoflagellates (Meier et al., 2007) and cyanobacteria (Merz-Preiß, 2000)), as well as our incomplete
understanding of their ecology and physiology and a lack of *in situ* global measurements. Despite recent advances

in understanding the biomass distribution of coccolithophores and foraminifera (O'Brien et al., 2013, 2016;
Schiebel and Movellan, 2012), and how these may relate to carbonate chemistry (e.g. Bach et al., 2015; Evans et
al., 2016), we still have very little idea of the relative magnitude (or biogeography) of their respective rates in
terms of production or export (e.g. Schiebel, 2002; Berelson et al., 2007).

A key misconception when considering oceanic $CaCO_3$ production by coccolithophores is the enigmatic role of
*Emiliania huxleyi* in satellite imagery of $CaCO_3$ (or particulate inorganic carbon, PIC). The characteristic light
scattering properties of PIC particles, the size of *E. huxleyi* coccoliths (Balch et al., 1996), in addition to its
ubiquitous distribution, tendency to shed excess coccoliths, and propensity to form massive turbid blooms, has set
the focus on this species in the development of algorithms for satellite ocean-colour remote sensing of PIC

measurements (Balch et al., 2005). Several studies have used satellite images to examine trends in global PIC
production, in terms of regional variability, areal magnitude (e.g. Balch et al., 2005, 2007; Freeman and



Lovenduski, 2015; Hopkins and Balch, 2018) and coccolithophore ecology (e.g. Hopkins et al., 2015). However, these budgets are likely to be less accurate in terms of fully accounting for PIC contributions from the whole coccolithophore assemblage due to their diversity in coccolith-specific backscattering coefficients (Balch et al.,

1999), which arise due to considerable diversity in coccolith sizes, shapes, morphologies and $CaCO_3$ contents (Young and Ziveri, 2000; Young et al., 2003). Relatively small differences in the $CaCO_3$ content of the various *E. huxleyi* morphotypes (Young et al., 2003; Poulton et al., 2011) can have significant impact in terms of $CaCO_3$ formation at the scale of mesoscale blooms (Poulton et al., 2013). Moreover, recent studies have highlighted the potential for less abundant, yet more heavily calcified species other than *E. huxleyi* to dominate coccolithophore

$CaCO_3$ production (Daniels et al., 2014, 2016), and hence there is a need to better consider community-wide $CaCO_3$ production. Satellites also detect relatively localised bloom events, whereas the non-bloom production in temperate waters may be relatively substantial (e.g. Poulton et al., 2010). Moreover, the areal extent of mid- to low-latitude waters confers them with a substantial global role in integrated $CaCO_3$ budgets (e.g. Balch et al., 2005; Marañón et al., 2016).


Here we focus on the pelagic $CaCO_3$ production (CP) from the global ocean, taking advantage of a recent increase in the oceanic measurement of their calcification rates across diverse ocean environments. As almost all coccolithophore species, with a few notable exceptions (Young et al., 1999), produce the calcite form of $CaCO_3$, the terms $CaCO_3$ production and calcite production may be considered interchangeable for coccolithophores.

However, it also has to be noted that the methodology (see Sect. 2.1.2) to determine CP does not distinguish the *actual* form of $CaCO_3$, whether it is calcite (coccolithophores, foraminifera, some dinoflagellates) or aragonite (foraminifera, pteropods, corals).

The ecology and physiology of coccolithophores has been reviewed numerous times (see Paasche, 2002;

Zondervan, 2007; Boyd et al., 2010; Raven and Crawfurd, 2012; Monteiro et al., 2016; Taylor et al., 2016). Recent advances also include a better understanding of coccolithophore calcification in the context of carbonate chemistry (Bach et al., 2015), energetic considerations (Monteiro et al., 2016) and phytoplankton succession (Hopkins et al., 2015). To date only two studies have previously collated and synthesised calcification rates across the global ocean (Balch et al., 2007; Poulton et al., 2007); however, there are now numerous studies published over the last

decade (see Table 1), which reformulates the global perspective on CP by coccolithophores. Poulton et al. (2007) previously noted a significant geographical bias in the data collected, with most data originating from (sub-)tropical waters, whereas measurements are now available from more diverse regions, such as the Arctic (e.g. Charalampopoulou et al., 2011; Balch et al., 2014; Daniels et al., 2016) and Southern Ocean (e.g. Balch et al., 2016; Charalampopoulou et al., 2016).


Paasche (1962, 1963) first proposed direct measurements of coccolithophore production of $CaCO_3$ by demonstrating that radioactive carbon-14 ([14]C) could trace the production of both organic (via photosynthesis) and inorganic carbon (via calcification) by coccolithophores in the laboratory. The use of [14]C to measure photosynthesis dates back to Steeman Nielsen in the 1950s (see Barber and Hilting, 2002), with a key step being

the acid-treatment of filtered material (post-incubation) to remove any remaining [14]C-labelled dissolved inorganic



carbon ($^{14}$C-DIC) as $^{14}$CO$_2$ (e.g. Knap et al., 1996; Marra, 2002). However, if the filtered samples are rinsed (extensively) with unlabelled seawater to remove any unfixed $^{14}$C-DIC before acid exposure, then the $^{14}$CO$_2$ liberated upon acidification of the filters represents $^{14}$C-DIC fixed into Ca$^{14}$CO$_3$ (i.e. CP).

Two techniques exist to utilise this production of $^{14}$CO$_2$ to measure calcification. The first requires filtering $^{14}$C-labelled samples post-incubation through two filters; one is then fumed with acid (e.g. hydrochloric acid) to remove the Ca$^{14}$CO$_3$ (and then termed particulate organic production), while the other is left un-fumed (termed total particulate production). Calcification (particulate inorganic production) then represents the difference between the particulate production of these two filters. This "difference" method was first used in culture

experiments (Paasche, 1963) and then later at sea by Balch et al. (1992) in the Gulf of Maine, while Fernández et al. (1993) used this technique to characterise CaCO$_3$ production within an extensive bloom of coccolithophores in the North Atlantic. The second method (the 'micro-diffusion technique', MDT) directly captures the $^{14}$CO$_2$ liberated from Ca$^{14}$CO$_3$, providing a direct measurement of calcification with a high degree of accuracy. The MDT was originally developed by Paasche and Brubak (1994) and modified by Balch et al. (2000) for ship-based

research, and has now been used in numerous field studies (see Table 1).

The objective of this study was to create a database compiling all the available *in situ* measurements of CaCO$_3$ production in the ocean. By synthesising the numerous individual datasets into one database, we hope to provide a baseline for validation of model outputs and satellite algorithms. Two previous data syntheses (Balch et al.,

2007; Poulton et al., 2007) were published around a decade ago, though the datasets included were smaller with some geographical biases (i.e. a large amount of (sub-)tropical data): the present dataset aims to synthesise *all* the available calcification rate data, and will be updated as new data becomes available.

## 2    Data and Methods

The database is available at PANGAEA as doi: 10.1594/PANGAEA.888182 (Poulton et al., 2018).

### 2.1    Database construction

#### 2.1.1    Database summary

Data were compiled from the available scientific literature, with permission to include each dataset acquired from

the lead author and/or principal investigator where appropriate. Following the initial data collection, oceanographic cruises with unpublished data were identified, and the data owners and originators contacted for permission and access to include those further datasets. The data consist of direct measurements of CaCO$_3$ production (CP) and primary production (PP), cell counts of coccolithophores (where available, not differentiated by species in this database), and ancillary data, including the collection date and year, latitude, longitude, sampling

and light depth (when available), incubation length ($\leq$12 or 24 h) and method of measuring CP (via 'difference' or MDT). The quality-controlled (see Sect. 2.2) database consists of 2765 data points, with coccolithophore cell counts matched to 1301 data points.



### 2.1.2 Calcium carbonate production and primary production

$CaCO_3$ production (CP) was mostly measured using $^{14}C$, with one study using $^{45}Ca$ as a tracer (Van der Wal et al., 1995) (Table 1). Water samples (<0.5 L) were collected via various methods (e.g. Go-Flo bottles, Niskin bottles with rosette samplers, uncontaminated surface seawater supply), spiked with various activities (~2 to 100 µCi or ~74 to 3700 kBq) of $^{14}C$-labelled bicarbonate and incubated for 6 to 24 h under various light regimes (see original references in Table 1 for full methodological details). As CP is measured on small volumes (<0.5 L), with

coccolithophore abundances ranging from 10 to 2000 cells $mL^{-1}$, such measurements are likely, but not exclusively, to exclude CP from large (63-200 µm) and rare calcifying organisms, such as foraminifera (typically ≤0.01 $mL^{-1}$ or ≤10 $L^{-1}$) or pteropods (typically ≤0.001 $mL^{-1}$ or ≤1 $L^{-1}$).

Two techniques were used with $^{14}C$: the 'difference' method and the MDT (Table 1). For measurements by

'difference', the incubations are terminated by filtering the sample onto two replicate filters. One filter is fumed with acid (most often hydrochloric acid) to remove the acid-labile inorganic carbon (i.e. $CaCO_3$), leaving non-acid labile particulate organic carbon, while the other is untreated. The radioactivity of the two filters is measured using liquid scintillation counting to determine the total carbon fixation (inorganic + organic carbon fixation, often termed total particulate production) on the untreated filter and the organic carbon fixation (often termed primary

production, PP) on the acid-fumed filter. $CaCO_3$ production is then determined as the difference between these two measurements. This technique can provide accurate estimates of CP when rates are high (and ratios of CP to PP are near unity), such as in coccolithophore blooms (e.g. Fernandez et al., 1993) or laboratory cultures (e.g. Balch et al., 1992). However, the accuracy of this technique suffers significantly in oceanic samples where CP can be much smaller than PP (less than a tenth of PP; Poulton et al., 2007), such that CP is calculated as the

difference between two large numbers with potentially large errors (see Appendix A).

The MDT overcomes the limitations of the difference method, as it is able to measure directly both CP and PP from the same water sample, using only one filter (Balch et al., 2000; Paasche and Brubak, 1994). Following the incubation of seawater spiked with $^{14}C$-bicarbonate, the sample is filtered and extensively rinsed with non-labelled

pre-filtered seawater, and the filter is placed into a glass vial. A glass fibre filter (e.g. Whatman GFA), pre-soaked with an alkaline solution (Balch et al., 2000) or β-phenylethylamine (Poulton et al., 2006; Balch et al., 2011) is suspended within the vial to act as a $CO_2$ trap. The sample filter is then acidified (e.g. 1% phosphoric acid; see Balch et al., 2000), liberating the acid-labile inorganic carbon ($CaCO_3$) as $CO_2$. The resultant $^{14}CO_2$ is captured on the glass fibre filter over time (>12 h), which is then moved to a fresh vial from which CP can be measured

directly. Measuring CP and PP from the same filter allows the MDT to reduce experiment error, resulting in more precise, reliable and accurate measurements of CP (Marañón and González, 1997; Balch et al., 2000, 2007). As a measure of abiotic isotope labelling of material, a formalin-killed blank incubation is run in parallel to the 'light' samples, and later subtracted (Balch et al., 2000).

An alternative method for measuring CP is through using $^{45}Ca$ as the tracer rather than $^{14}C$ (Van der Wal et al., 1995). Seawater is incubated with $^{45}CaCl$ and subsequently filtered. The advantage of this method is that it does not require the separation of inorganic and organic uptake, as required for either $^{14}C$ technique. However, $^{45}Ca$



forms strong ionic bonds, such that unincorporated $^{45}$Ca is not easily removed by rinsing and blanks are often large (Balch et al., 2007; Van der Wal et al., 1995).


With the ability to measure low rates of CP, the MDT is the currently preferred method for measuring CP in the ocean, compared to both the "difference" method and $^{45}$Ca. This is reflected in the database, where 2527 (91.4 %) of the data points were measured using the MDT, 215 (7.8 %) using the difference technique, and 23 (0.8 %) using $^{45}$Ca. For a comparison of the performance of the MDT and the difference technique on oceanic

coccolithophore communities see Appendix A.

The majority of the data in the current database comes from 24 h incubations, which captures a complete daily cycle of growth, and accounts for any CP (or loss of fixed carbon via mortality) occurring at night (Poulton et al., 2007, 2010). However, several earlier studies used shorter incubation lengths, and are highlighted in Table 1. The

measurements collected by Poulton et al. (2006, 2007) were only incubated over the local daylight period (10-16 h), and it was assumed that negligible CP occurred at night (e.g. Linschooten et al., 1991; but see Paasche, 1966; Balch et al., 1992). Samples collected in the Gulf of Maine (Balch et al., 2008) were brought back to the laboratory to measure photosynthesis and CP in half-day, 'CalCOFI-style' (California Cooperative Oceanic Fisheries Investigations) incubations (see Mantyla et al., 1995). The half-day incubations minimised bottle effects (Balch

et al., 2008), ran from local apparent midnight to midday, and were converted to daily rates of CP using ratios of 12 and 24 h incubations. Finally, Lam et al. (2001) incubated for 5 h around midday and calculated an hourly rate of CP. In this database, this hourly rate has been scaled up by the calculated day-length based on latitude, longitude and seasonal timing of the study (see Kirk, 1994), assuming that no dark calcification occurred, although this has been observed in laboratory cultures (Paasche, 1966; Linschooten et al., 1991; Balch et al., 1992). When

appropriate, CP and PP data were standardised into units of mmol C m$^{-3}$ d$^{-1}$.

### 2.1.3    Cell counts

When available, cell counts for coccolithophores (Table 1) were generally performed using either polarised light microscopy (Balch et al., 2004, 2008, 2012; Balch et al., unpublished; Daniels et al., 2016; Mayers et al., 2018;

Mayers et al., unpublished; Poulton et al., 2010, 2013, 2014; Poulton et al., unpublished) or scanning electron microscopy (Charalampopoulou et al., 2011, 2016; Loucaides et al., unpublished; Poulton et al., unpublished). Other methods for cell counting in the database include inverted light microscopy of formalin-preserved samples (Lipsen et al., 2007; Marañón et al., 2016) and use of a haemocytometer (Marañón and González, 1997).

With the exception of Marañón and González (1997), who report only the concentration of *E. huxleyi*, cell counts correspond to the total concentration of coccolithophores in each water sample. In 556 of these samples, both the total coccolithophore abundance and the *E. huxleyi* abundance are reported (Charalampopoulou et al., 2011, 2016; Daniels et al., 2016; Lipsen et al., 2007; Loucaides et al., unpublished; Mayers et al., 2018.; Poulton et al., 2010, 2013; Poulton et al. unpublished). In all cases, the counts are reported in cells mL$^{-1}$.






### 2.1.4 Optical depths, depth integration and surface data

There are 314 vertical profiles of CP within the database presented. From these profiles, depth-integrated values were calculated representing euphotic zone integrated CP (in which the euphotic zone is taken as either 1% of incident irradiance (e.g. Poulton et al., 2006) or 0.1% (e.g. Balch et al., 2011) in the different studies. Herein, it is

assumed that CP only occurs within the euphotic zone and, therefore, euphotic zone integrated CP represents total water column CP by coccolithophores. Though coccolithophores may occur considerably deeper than the 1% irradiance depth (see e.g. Poulton et al., 2017), integration to the base of the euphotic zone allows comparison with other water-column processes frequently integrated to this depth (e.g. primary production, new production).

The light levels of the sampling depths, as a percentage of incident PAR, were provided either by the data originators or taken from the corresponding literature. Light depths were then converted to an equivalent optical depth by taking the negative natural logarithm approach where the 1% incident irradiance depth has an optical depth of 4.6 (see Kirk, 1994). The profiles were integrated by linearly interpolating using the sampling depths. There are 314 unique sampling stations with enough vertical resolution ($n \geq 4$) to calculate euphotic zone integrals

for CP (and PP) within the database.

However, a number of datasets included only upper ocean sampling and a subset of surface of data was created by extracting data collected from less than 20 m. In cases where multiple measurements were collected in this shallow window, only the data collected from the uppermost depth were extracted for the surface data comparison.


## 2.2 Log normal distribution and quality control

Rates of CP and the abundance of coccolithophores in the ocean can range from zero when coccolithophores are either completely absent (e.g. in high latitude polar waters) or below the limit of detection, up to extremely high values that may occur, for example, in a coccolithophore bloom. Consequently, both the CP rates and cell

abundances can vary over many orders of magnitude, exhibiting a log-normal distribution (Fig. 4A) when excluding zero-value data. This distribution is typical of many biological processes (Limpert et al., 2001). For log-normally distributed data the geometric mean, rather than arithmetic mean, best characterises the data and hence we report only the geometric means from the database.

We quality-controlled the datasets by first removing all negative CP values. Negative values can occur in the difference method as CP is significantly smaller than PP and when the variability (replication) in PP between filters can be greater than the CP signal (see Appendix A). A negative rate can also be obtained using the MDT if the formalin-killed blank is greater than the measured rates, as may occur at low light levels at the base of the euphotic zone (e.g. Poulton et al., 2010) or in water samples with low rates of CP. A negative rate of CP cannot

actually occur using the (single point) radioisotope tracer technique and, therefore, these rates were eliminated from the database. The decision was also made to remove all zero-value data points of CP and cell counts. In general, the methods used to measure CP and cell abundances are not sensitive enough to distinguish between true zero values and those below their limit of detection. Furthermore, the limit of detection will vary between



users and specific details of their methods (e.g. volume used, spike activity added), and hence it is more consistent
to remove all zero-values from the database rather than set an arbitrary limit of detection for the whole database.

## 3 Results and Discussion

### 3.1 Data distribution

Figure 1 shows the spatial distribution of the database of CP. The Atlantic Ocean has the best data coverage,
particularly in the high latitudes of the North Atlantic. Coverage of the Southern Ocean is constrained to the
Atlantic and Indian Sectors. The Pacific Ocean is poorly represented with no coverage in the Western Pacific.
Although there is a large number of data in the Indian Ocean, it is restricted to the Arabian Sea (Balch et al., 2000).
The most heavily sampled region is the Gulf of Maine (Table 1) (Balch et al., 2008, 2012).

There are significant gaps in the spatial distribution of the dataset, with a particular bias towards the Atlantic
Ocean. However, the spatial coverage has greatly increased since 2006 (see Balch et al., 2007; Poulton et al.,
2007), particularly in the high latitudes. Figure 2 shows the temporal and seasonal distribution of the data. The
increase in spatial coverage is partly attributable to the general increase in data collection, with 44% of the data
collected since 2006. However, the seasonal distributions demonstrate bias towards the summer months of the
Northern (June – August) and Southern (December – February) hemispheres (Figs. 2B and 2C).

Figure 3 shows the vertical distribution of the database, in terms of depth and optical depth. Most data were
collected from relatively shallow waters: 60% of samples were collected from less than 20 m, and 41% at more
than 50% of surface irradiance (optical depths <0.7).


### 3.2 Magnitude of $CaCO_3$ production rates

The entire dataset of CP is well approximated by a log-normal distribution (Fig. 4A), with a geometric mean of
16.1 µmol C m$^{-3}$ d$^{-1}$. The total range in CP is from 0.01 to 8398 µmol C m$^{-3}$ d$^{-1}$, which has greatly expanded
compared to Poulton et al. (2007). The highest measured CP rate occurred in the Gulf of Maine in July 2002
(Balch et al., 2012). Rates of CP in excess of 5000 µmol C m$^{-3}$ d$^{-1}$ were measured twice in a coccolithophore
bloom in the Celtic Sea in April 2015 (Mayers et al., 2018). In total, there are 23 occurrences of CP rates over
1000 µmol m$^{-3}$ d$^{-1}$, very likely indicative of coccolithophore blooms (Poulton et al., 2007, 2013).

#### 3.2.1 Surface $CaCO_3$ production

The surface CP data are also approximated by a log-normal distribution (Fig. 4B), with a slightly higher geometric
mean (20.3 µmol C m$^{-3}$ d$^{-1}$) than the complete dataset. Surface CP spans the entire range in CP (0.01 – 8398 µmol
C m$^{-3}$ d$^{-1}$), and is highly variable in the ocean (Fig. 5A). In general, surface CP is higher in the high latitude North
Atlantic (Fernández et al., 1993; Poulton et al., 2010; Daniels et al., 2016), the Patagonian Shelf region of the
South Atlantic (Poulton et al., 2013), the North Pacific (Lipsen et al., 2007) and in the Arabian Sea (Balch et al.,
2000). Some of the lowest rates of CP are observed in the Southern Ocean (Charalampopoulou et al., 2016),
although there is no clear pattern in the global distribution. The higher CP rates tend to be in well-sampled regions



as studies have targeted areas known or predicted to be areas of significant coccolithophore abundances. This geographical (and seasonal) sampling bias may have resulted in an inflated global mean value of CP as there are only a few data points from regions where coccolithophores are thought to be rare (e.g. the subtropical Pacific and

high latitude polar seas).

### 3.2.2    Integrated CaCO₃ production

Integrated CP is also log-normally distributed, with a geometric mean of 1.19 mmol C $m^{-2}$ $d^{-1}$, and a range of <0.01 to 6 mmol C $m^{-2}$ $d^{-1}$. As there are significantly less vertical profiles of CP (314) than surface measurements

of CP (1103), the spatial coverage of integrated CP is much sparser (Fig. 5B), particularly in the high latitude North Atlantic. The pattern of integrated CP is slightly different to that of surface CP. Although integrated CP is high on the Patagonian Shelf, in the Arabian Sea and in the sub-polar North Atlantic, it is also high in the Equatorial Pacific. This partly reflects the deeper euphotic zones (> 60 m) in the Equatorial Pacific compared to the sub-polar regions (< 50 m) (see Landry et al., 2011). The vertical distribution of CP against optical depth is

shown in Fig. 4C. The lack of relationship between CP and optical depth for the entire dataset is partly due to the fact that the global variation in CP for any optical depth is greater than the vertical pattern in CP.

There is a strong positive correlation between surface CP and integrated CP (Pearson's product-moment correlation, $r = 0.83$, $p < 0.001$, $n = 314$), when the logarithms of both are taken (Fig. 6). While a strong correlation

between surface PP and integrated PP has been previously observed (e.g. Poulton et al., 2007), and is observed here (Fig. 7A), the relationship observed for CP by Poulton et al. (2007) was statistically weaker ($r = 0.47$, $p < 0.001$, $n = 68$). This difference may relate to the greater degree of temperate data in the larger database, where light will be a strong driver of deep CP within the mixed layer. In contrast, the previous database had a greater degree of tropical data, where deep thermocline CP may be strongly light-limited and/or dependent on non-

autotrophic nutrition (Poulton et al., 2017).

### 3.3    CaCO₃ production versus primary production

The ratio of CP to PP is highly variable in the database (Fig. 7A), with a log-normal distribution (Fig. 7B). The average (geometric mean) ratio of CP:PP for the total database is 0.02, though it has a range from as low as below

0.0001 to as high as over 5. This distribution is highly similar to that observed by Poulton et al. (2007), though there is a much greater degree of variability within the expanded dataset (and potential issues with the more extreme values).

Broadly similar trends are observed when considering both surface CP and PP (Fig. 7C) and integrated CP and

PP (Fig. 7E), with average CP:PP around 0.01 and 0.03, respectively. As the average CP:PP ratio is lower in surface waters than in the total dataset, there may be a decoupling of PP and CP with depth and a greater light-dependency for photosynthesis than calcification (see Balch and Kilpatrick, 1996; Balch et al., 2000, 2011; Poulton et al., 2007, 2010). The effect of optical depth on the ratio of CP to PP is shown in Fig. 7D. No general trend is identifiable, with data from deeper optical depths having similar CP:PP ratios to surface values. CP at

depths below the light levels required for photosynthesis may also relate to non-autotrophic nutritional strategies





by deep-dwelling coccolithophore species (e.g. Poulton et al., 2017). No clear relationship is found between latitude and CP:PP (Fig. 7F).

The log-normal relationship between CP and PP can be potentially useful in a practical sense. Oceanic rates of PP are much more widely measured in field programmes, and therefore PP is better constrained than CP. By using the log-normal relationship between CP and PP identified in the global database we may be able to gain greater insights into spatial and temporal patterns, as well as the extent of CP. For example, global marine PP is estimated to be ~50 Gt C yr$^{-1}$ (Field et al., 1998) while global CP is poorly constrained with estimates ranging from 0.4 to 8 Gt C yr$^{-1}$ (Balch et al., 2007; Berelson et al., 2007). A first order approximation of global CP, using the average 360 CP:PP of 0.02 from the database, gives an estimate of ~1 Gt C yr$^{-1}$. This value is only slightly lower than a recent estimate, based on coccolithophore ecophysiology, of 1.42 Gt C yr$^{-1}$ by Hopkins and Balch (2018). Clearly, more sophisticated methods can also be used in the future with the global CP database to better approximate regional and global estimates of CP.

**3.4    Cell-normalised calcification**

A key consideration in measurements of oceanic biogeochemical rates is the accuracy and representativeness of the resulting values. For PP (and photosynthesis), normalising rates to concentrations of chlorophyll-*a* (or phytoplankton carbon) gives information about the variability in production per unit biomass, where a solid understanding of photo-physiology (e.g. Behrenfeld and Falkowski, 1997; Falkowski and Raven, 1997) helps to 370 identify physiologically unrealistic rates. In the case of CP, normalising to chlorophyll-*a* or particulate inorganic carbon (PIC) can be considered inappropriate, as neither of them fully represent living coccolithophore biomass (Poulton et al., 2007).

We suggest that a more physiologically sound approach is to normalise CP to coccolithophore cell abundance 375 (Poulton et al., 2010; Fig. 8), which provides a measure of calcification per unit 'biomass' (cell-CP) comparable (in basic terms) to chlorophyll-normalised photosynthetic rates. Figure 8 shows the variability in cell-CP when normalising CP to total coccolithophore abundance using the matched values available in the database.

Within natural coccolithophore communities, CP is dependent on cell abundance, species composition and the 380 rate of calcification per cell (Poulton et al., 2010; Daniels et al., 2014). Using cell-CP to examine coccolithophore dynamics is particularly appropriate when applied to communities dominated by a few species due to the sensitivity of cell-CP to cellular CaCO$_3$ content, and hence species composition (Poulton et al., 2010; Charalampopoulou et al., 2011). More recently it has also been modified to account for variability in growth rates and species composition, allowing species-specific contributions to community CP to be constrained (Daniels et 385 al., 2016).

The values of cell-CP in Fig. 8 range from <0.001 to 46.4 pmol C cell$^{-1}$ d$^{-1}$, with a geometric mean of 0.42 pmol C cell$^{-1}$ d$^{-1}$ ($n$ = 1272). The cell-CP of *E. huxleyi* dominated natural communities is known to be variable, with




average reported field values ranging from 0.16 to 0.65 pmol C cell$^{-1}$ d$^{-1}$ across non-bloom communities in the

North Atlantic and Southern Ocean, as well as bloom communities on the Patagonian Shelf (Poulton et al., 2010, 2013; Charalampopoulou et al., 2016). A cell-CP of 0.023 pmol C cell$^{-1}$ d$^{-1}$, equivalent to an *E. huxleyi* coccolith production rate of ~1 day$^{-1}$ (Young and Ziveri, 2000; Poulton et al., 2010), can be considered close to a theoretical minimum cell-CP for *E. huxleyi*. Thus, samples in Fig. 8 with a cell-CP lower than this value are likely to be dominated by coccolithophore species with much lower cellular CaCO$_3$ contents than *E. huxleyi* (e.g.

*Calciopappus caudatus*; Daniels et al., 2016; Mayers et al., 2018). Conversely, those samples with a cell-CP significantly greater than 1 pmol C cell$^{-1}$ d$^{-1}$ are likely to be dominated by coccolithophore species with greater cellular contents than *E. huxleyi*. Cell-CP for heavily calcified coccolithophore species such as *Coccolithus pelagicus*, may reach as high as ~8.3 pmol cell$^{-1}$ d$^{-1}$ or ~23.2 pmol C cell$^{-1}$ d$^{-1}$ for *C. braarudii* (depending on the cell CaCO$_3$ content and growth rate; Daniels et al., 2014). A theoretical maximum could therefore be considered

as ~40 pmol C cell$^{-1}$ d$^{-1}$; based, for example, on a maximum growth rate of 0.6 d$^{-1}$ for the heaviest extant coccolithophore species (*Scyphosphaera apsteinii*; Young and Ziveri, 2000) with ~10 to 12 coccoliths per coccosphere (Young et al., 2003) and a cell CaCO$_3$ of ~54 to 65 pmol C cell$^{-1}$.

The values of cell-CP in the CP database (Fig. 8) are mostly within these theoretical limits, indicating that they

can be viewed as 'realistic' in the context of physiological limitations (growth and coccolith production rates) and extant species composition (cell and coccolith calcite quotas). Hence, cell-CP provides a useful benchmark for examining the physiological and growth dynamics of coccolithophore communities (e.g. Poulton et al., 2010, 2013; Charalampopoulou et al., 2016; Daniels et al., 2016; Mayers et al., 2018), as well as acting as a reality check for oceanic measurements of CP. The relative species composition of mixed communities also has to be considered

when examining trends in cell-CP (and total CP), which fittingly links together the biogeochemically important role of coccolithophores in CP with their diversity in form, function and ecophysiology.

## 4 Conclusions and future recommendations

We have assembled a database of 2765 data points of CP across the global ocean (from oceanic to coastal and

tropical to polar), resulting in a significant increase in both the size and the spatial coverage of previous syntheses of similar measurements (Balch et al., 2007; Poulton et al., 2007). This database may be valuable in global-scale studies of CP and coccolithophores, though the main limitations of the current database are its spatial coverage, with particularly poor coverage in the Pacific, and a significant temporal bias towards spring/summer sampling.

The CP data are log-normally distributed, such that geometric means are required for examining the CP data. There is significant variability in the CP data, with no clear patterns in the global distribution of either surface or integrated CP, although there is a strong relationship between surface CP and integrated CP. We recommend that future field studies of CP use the MDT technique in combination with cell counts to obtain estimates of cell-CP to 'ground-truth' the CP measurements. Cell-CP also provides further insights into the coccolithophore diversity

and physiology underpinning measurements of CP. The MDT technique is the only direct method capable of accurately measuring low rates of CP (see Appendix A). The CP database is freely available and stored permanently at PANGAEA (Poulton et al., 2018; doi: 10.1594/PANGAEA.888182) and there are plans to update



it as and when new data becomes available. We hope that the database will be useful in model and satellite validation, and for examining spatial and temporal variability in CP on a global scale.


**Author contributions.** CJD and AJP devised the data synthesis and collated the data, with earlier assistance from RH. All authors contributed data to the database. WMB and EM commented on early drafts of the paper. All authors commented on subsequent drafts. KMJM and JH also assisted with the final versions of the figures.

**Competing interests.** The authors declare that they have no conflicts of interest.

**Acknowledgements.** The authors wish to thank the research scientists, technicians, students and crew who contributed to the collection of these data. The authors also recognise funding from the UK Natural Environmental Research Council (NERC), the US National Science Foundation (NSF), National Aeronautics and Space
Administration (NASA), and the Spanish Ministry of Science and Innovation.

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





**Table 1: Data Sources**

| Principal Investigator | Number of CP measurements | | Region | Method | Incubation length (hrs) | Cell count method | Reference(s) |
|---|---|---|---|---|---|---|---|
| | **Depth Integrals** | **Total Points** | | | | | |
| Balch | - | 1 | Gulf of Maine | Difference | 24 | | Balch et al. (1992) |
| Balch | 11 | 70 | Equatorial Pacific | Difference | 24 | | Balch and Kilpatrick (1996) |
| Balch | 51 | 508 | Arabian Sea | MDT | 24 | | Balch et al. (2000) |
| Balch | - | 492 | Gulf of Maine | MDT | ~12 | Light microscopy | Balch et al. (2004); Balch et al. (2008); Balch et al. (2012) |
| Balch | 28 | 157 | Equatorial Pacific | MDT | 24 | | Balch et al. (2011) |
| Balch | 29 | 153 | Southern Ocean: Atlantic | MDT | 24 | Light microscopy | Balch et al. (2016) |
| Balch | 28 | 145 | Southern Ocean: Indian | MDT | 24 | Light microscopy | Balch et al. (2016) |
| Charalampopoulou | 6 | 32 | Arctic Ocean | MDT | 24 | Scanning Electron Microscopy | Charalampopoulou et al. (2011) |
| Charalampopoulou | 6 | 59 | Southern Ocean: Atlantic | MDT | 24 | Scanning Electron Microscopy | Charalampopoulou et al. (2016) |
| Crawford & Lipsen | 4 | 21 | Sub-polar North Pacific | MDT | 24 | | Unpublished |
| Daniels | - | 24 | Arctic Ocean | MDT | 24 | Light microscopy | Daniels et al. (2016) |
| Daniels & Fragoso | - | 22 | Southern Ocean: Atlantic | MDT | 24 | | Unpublished |
| Feng | - | 1 | Sub-polar North Atlantic | MDT | 24 | | Feng et al. (2009) |
| Fernandez | 5 | 28 | Sub-polar North Atlantic | Difference | 24 | | Fernandez et al. (1993) |
| Fernandez | 4 | 22 | Norwegian Fjord | Difference | 24 | | Fernandez et al. (1996) |
| Lam | - | 1 | Sub-polar North Pacific | MDT | 5 | | Lam et al. (2001) |
| Lipsen | 38 | 219 | Sub-polar North Pacific | MDT | 24 | Inverted Light Microscopy | Lipsen et al. (2007) |
| Loucaides | - | 18 | Arctic Ocean | MDT | 24 | Scanning Electron Microscopy | Unpublished |
| Marañón | 2 | 40 | North Sea | Diff | 24 | Haemoyctometer | Marañón and Gonzalez (1997) |
| Marañón | 1 | 85 | Tropical Atlantic, Pacific and Indian Oceans | MDT | 24 | Inverted Light Microscopy | Marañón et al. (2016) |
| Marchetti | 3 | 21 | Sub-polar North Pacific | MDT | 24 | | Marchetti et al. (2006) |
| Mayers | 6 | 32 | Celtic Sea | MDT | 24 | Light microscopy | Unpublished |
| Mayers | 8 | 52 | Celtic Sea | MDT | 24 | Light microscopy | Mayers et al. (2018) |
| Mayers | 7 | 42 | Celtic Sea | MDT | 24 | Light microscopy | Unpublished |
| Poulton | 10 | 55 | Sub-tropical Atlantic | MDT | 10-16 | | Poulton et al. (2006) |
| Poulton | 11 | 70 | Sub-polar North Atlantic | MDT | 24 | Light microscopy | Poulton et al. (2010) |
| Poulton | 25 | 150 | Patagonian Shelf | MDT | 24 | Light microscopy | Poulton et al. (2013) |
| Poulton | 14 | 70 | Northwest European Shelf | MDT | 24 | Light microscopy | Poulton et al. (2014) |
| Poulton | - | 17 | Sub-polar North Atlantic | MDT | 24 | Scanning Electron Microscopy | Unpublished |
| Poulton & Adey | - | 53 | Sub-tropical Atlantic | MDT | 10-16 | | Poulton et al. (2007) |
| Rees | 9 | 54 | North Sea | Diff | 24 | | Rees et al. (2002) |
| Tynan | 4 | 28 | Arctic Ocean | MDT | 24 | | Unpublished |
| Van der Wal | 4 | 23 | North Sea | Ca-45 | 24 | | Van der Wal et al. (1995) |
| **Total** | **314** | **2765** | | | | | |





**Figures**

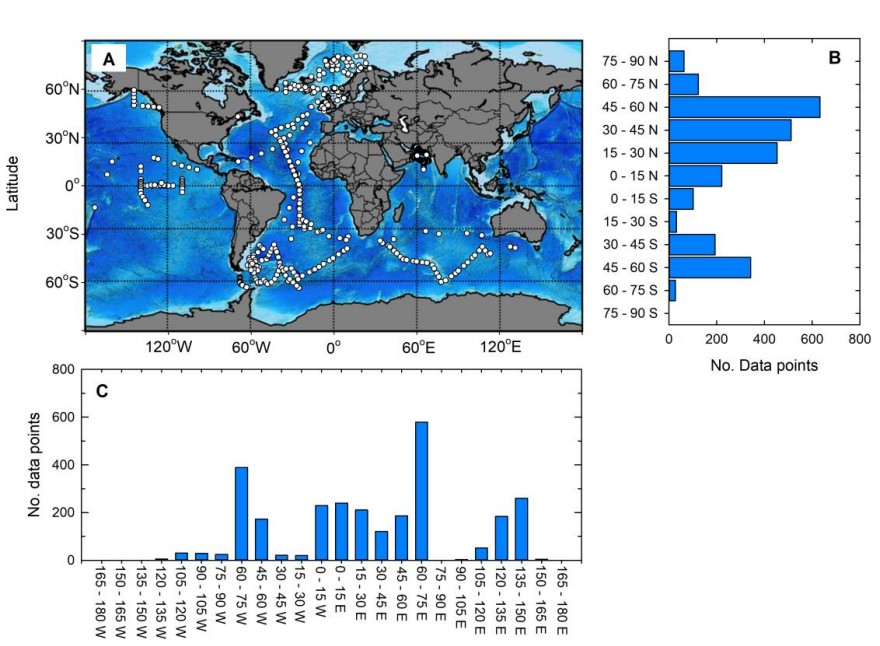


**Figure 1: Global map of CaCO₃ production data (A) and the frequency of data by latitude (B) and longitude (C). Global map in (A) superimposed on ocean bathymetry.**




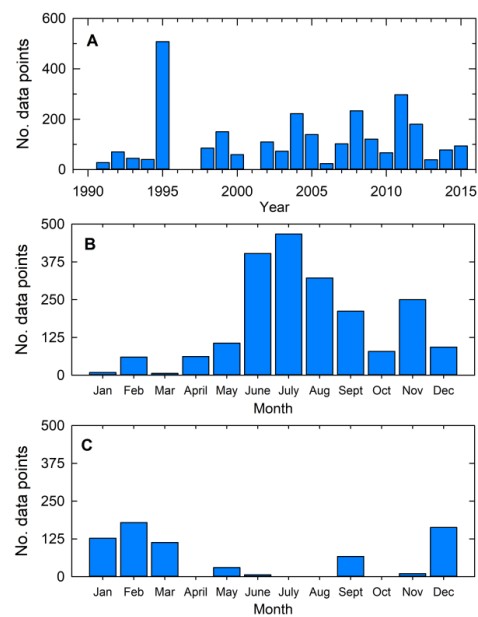

**Figure 2: Frequency of CaCO₃ production data by: (A) year of measurement; (B) month of measurement in the Northern Hemisphere; and (C) month of measurement in the Southern Hemisphere.**





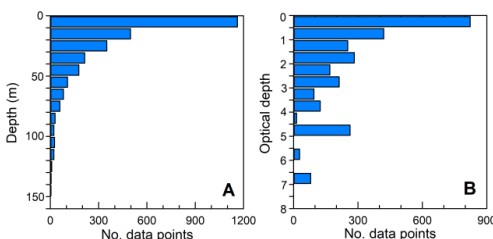

**Figure 3: Frequency of CaCO₃ production data by (A) sampling depth, and (B) optical depth. Depths relate to depth of sample collection, not incubation depth.**




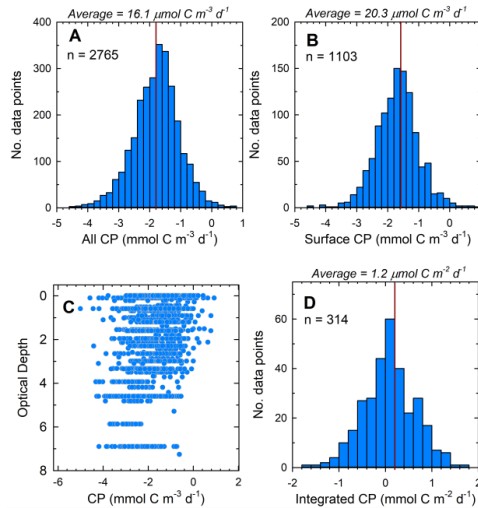

**Figure 4: Characteristics of the CaCO₃ production (CP) database: (A) measurement frequency versus all CP data; (B) measurement frequency for surface CP data only; (C) all CP data versus optical depth; and (D) measurement frequency for euphotic zone integrated CP. Panels (A), (B) and (D) have geometric means presented.**





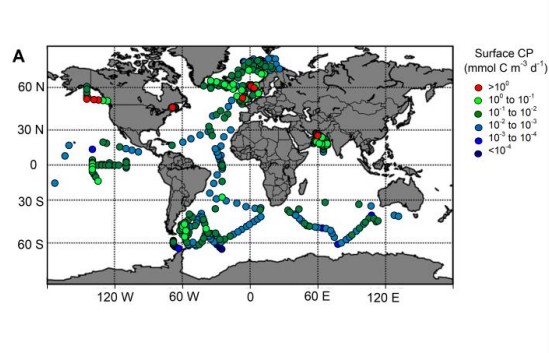

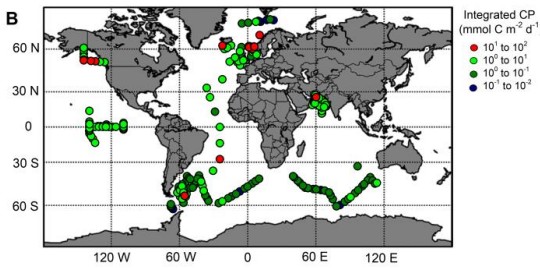


**Figure 5: Global maps of (A) surface CaCO₃ production (CP), and (B) euphotic zone integrated CP. Global maps superimposed on ocean bathymetry as in Fig. 1A.**




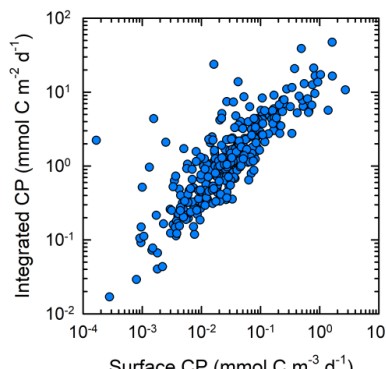

**Figure 6: Scatterplot of surface (<20 m) CaCO₃ production (CP) and euphotic zone integrated CP. The relationship between the two is statistically significant (Pearson's product-moment correlation, *r* = 0.83, *p* < 0.001, *n* = 314).**




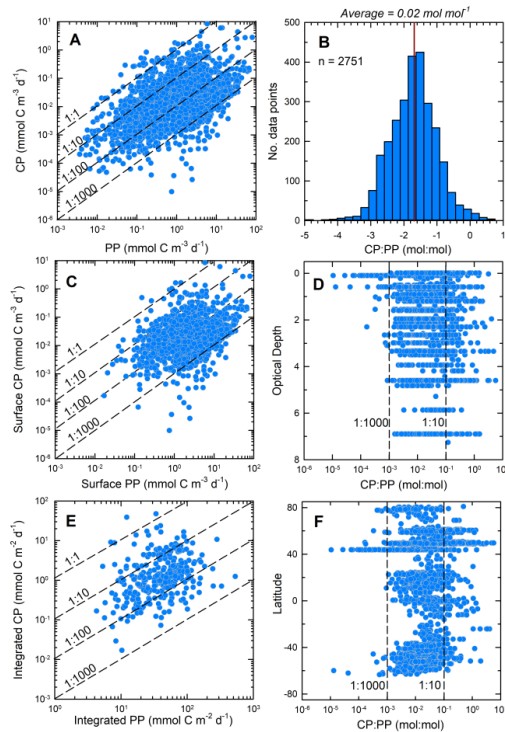

**Figure 7: Characteristics of the relationship between CaCO₃ production (CP) and Primary production (PP): (A) scatterplot of all CP and PP data; (B) frequency histogram of CP to PP ratios for all data; (C) scatterplot of only surface (<20 m) CP and PP; (D) scatter plot of CP:PP ratios against optical depth; (E) scatterplot of euphotic zone integrals of CP and PP; and (F) scatter plot of CP:PP ratios by latitude. Panels (A, C, D, E, F) include dashed lines of constant CP:PP. Panel (B) has the geometric mean ratio of CP to PP for all data indicated.**







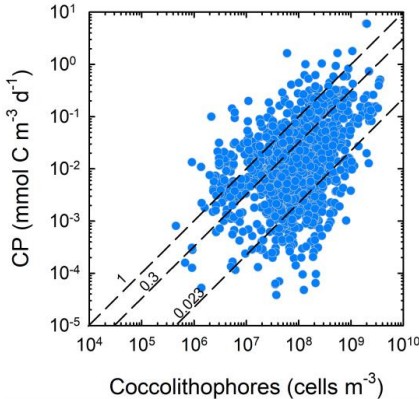

**Figure 8: Scatterplot of coccolithophore cell abundances and CaCO₃ production (CP) for all samples with matched count and rate data. Dashed lines indicate representative lines of cell-specific calcification (see Sect. 3.4).**




**Appendix A**

In summer 2010, during a research cruise to the North Atlantic (*RRS Discovery*, 4th July to 11th August, D354) seawater samples were collected and analysed using both the difference technique (Diff) and the Micro-Diffusion

Technique (MDT). Seawater for these comparisons was collected from sites ($n$ = 19) in the Iceland and Irminger basins from sub-surface (~5 m) waters during pre-dawn (06:00-07:00 h local time) deployments of a titanium CTD fitted with 10 L Niskin bottles on a rosette sampler.

For the MDT, 150 mL water samples (3 light, 1 formalin-killed) were spiked with 25 to 56 µCi (925-2,072 kBq)

of $^{14}$C-labelled sodium bicarbonate (Perkin-Elmer, UK) and incubated in on-deck incubators chilled with sea-surface seawater and with irradiance levels replicating ~30 to 40% of surface incidental irradiance using misty blue light filters (Lee Filters$^{TM}$, UK). Incubations were terminated after 24 h by filtering through 25 mm 0.2 µm polycarbonate filters, with extensive rinsing with fresh filtered seawater to remove any labelled $^{14}$C-DIC. Full methodology followed Poulton et al. (2010, 2013, 2014) and gave measurements of primary production (PP$_{MDT}$)

and CaCO$_3$ production (CP$_{MDT}$). The average coefficient of variation of triplicate (light) PP$_{MDT}$ measurements was 2% (10 to 28%) and 19% (1 to 72%) for CP$_{MDT}$, across a range of PP$_{MDT}$ from 1.5 to 5.3 mmol C m$^{-3}$ d$^{-1}$.

In parallel to the MDT measurements, measurements were also made of total particulate production (TPP) and primary production (PP$_{Diff}$), with the difference between the two being taken as CaCO$_3$ production (i.e., CP$_{Diff}$ =

TPP-PP$_{Diff}$), following the general methodology of Fernández et al. (1993) and Balch et al. (2000). Two slightly different protocols were used: for five experiments, TPP and PP$_{Diff}$ were measured from separate bottles, while for fourteen experiments, TPP and PP$_{Diff}$ were measured from the same bottle. Formalin-killed blanks were prepared in only seven experiments, with formalin values averaged and the average applied across the other twelve experiments. (Note: formalin-killed blank values were, on average, only ~4% of TPP and PP$_{Diff}$ values (range 2%

to 6% for both)).

Water samples (150 mL, 3 or 6 or 7) were collected, spiked with 3 to 13 µCi (108 to 489 kBq) of $^{14}$C-labelled sodium bicarbonate (Perkin-Elmer, UK) and incubated in parallel to the MDT samples. Incubations were terminated after 24 h with filtering through 25 mm 0.2 µm polycarbonate filters, with extensive rinsing with fresh

filtered seawater to remove any labelled $^{14}$C-DIC. Filters for the measurement of TPP were placed directly into scintillation cocktail after air-drying, while filters for PP$_{Diff}$ were either acid-fumed (hydrochloric acid, 2-3 h) or had 1 mL of 1% phosphoric acid added (20-24 h). The average coefficient of variation of the triplicate TPP measurements was 13% (1 to 27%) and 15% (4 to 21%) for PP$_{Diff}$, across a range of PP$_{Diff}$ from 1.4 to 2.9 mmol C m$^{-3}$ d$^{-1}$. (Note. The standard errors on the triplicate TPP measurements range from 95 to 1294 µmol C m$^{-3}$ d$^{-1}$,

while the standard errors for PP$_{Diff}$ range from 31 to 804 µmol C m$^{-3}$ d$^{-1}$; these values are comparable to the higher end of CP measured in the open-ocean, see main paper).

Comparison of TPP and PP$_{Diff}$ (Fig. S1A) showed that the two are significantly positively correlated ($r$ = 0.89, $p$ < 0.001, $n$ = 15), though PP$_{Diff}$ tended to be, on average, ~27% (5 to 54%) lower than TPP. PP$_{Diff}$ and PP$_{MDT}$ are

also closely correlated (Fig. S1B; $r$ = 0.71, $p$ < 0.005) with the average difference being only ~7% (although



differences did span -27% to 45%). However, TPP being around a third higher than $PP_{Diff}$ actually implies that rates of $CP_{Diff}$ (= TPP - $PP_{Diff}$) range from 164 to 2081 µmol C m$^{-3}$ d$^{-1}$ (Fig. S1C), with a cruise average of 952 µmol C m$^{-3}$ d$^{-1}$. In contrast, $CP_{MDT}$ only ranged from 4.1 to 141.8 µmol C m$^{-3}$ d$^{-1}$ (with a cruise average of 68 µmol C m$^{-3}$ d$^{-1}$), which is ~60 to ~9000 times *lower* than $CP_{DIFF}$ (Fig. S1C), though the two are significantly correlated

($r = 0.69$, $p < 0.005$, $n = 15$). Since $PP_{Diff}$ and $PP_{MDT}$ are strongly correlated, with a low relative difference between the two, the discrepancy between $CP_{Diff}$ and $CP_{MDT}$ derives from the much higher measurement of TPP. At this time there are no clear explanations for why TPP is so high relative to PP. It may be speculated that it is linked to the treatment of the samples (air-drying), as both PP measures are exposed to acid, and hence an unidentified source of labelled-carbon may be included in the TPP measurement but not those of PP. Further comment is

outside the scope of this study.

Objectively determining which CP measurement is accurate is not straightforward. One way is to consider the cell-normalised rates of calcification (cell-CP); i.e. which set of CP gives physiologically realistic cell-CP? For example, based on culture and field data *Emiliania huxleyi* may have cell-CP of 0.1 to 1.0 pmol C cell d$^{-1}$ (see

discussion in Poulton et al., 2010, 2013 and references therein, also Daniels et al., 2014). Maximum cell-CP for heavier species such as *Coccolithus pelagicus* may reach as high as ~8.3 pmol cell$^{-1}$ d$^{-1}$ or ~23.2 pmol C cell$^{-1}$ d$^{-1}$ for *C. braarudii* (depending on cell calcite and growth rates; Daniels et al., 2014). For the 2010 North Atlantic data, calculated cell-CP ($n = 12$) for $CP_{Diff}$ gives a range of cell-CP of 0.5 to 55.2 pmol C cell$^{-1}$ d$^{-1}$ (geometric mean = 4.40 pmol C cell$^{-1}$ d$^{-1}$), while for $CP_{MDT}$ a lower range of 0.02 to 3.36 pmol C cell$^{-1}$ d$^{-1}$ (geometric mean

= 0.25 pmol C cell$^{-1}$ d$^{-1}$) is calculated. Generally, cell-CP from $CP_{Diff}$ is on average 32 times higher (full range 7-206) than cell-CP calculated from $CP_{MDT}$. Cell-CP from the MDT gives values more in line with similar studies in the literature, although clearly further details of the species composition of the community (relative abundance, calcite content and growth rates) is required to fully reconcile the different estimates of cell-CF.

To conclude, the MDT provided CP and cell-CP rates which are fully consistent with the database and wider literature, whereas the Diff technique provides CP rates which are much higher than those most often found in non-bloom conditions in the North Atlantic and cell-CP rates which are high. Based on these observations, we suggest that the MDT is used for further field studies and the Diff technique is reserved for culture-based studies.






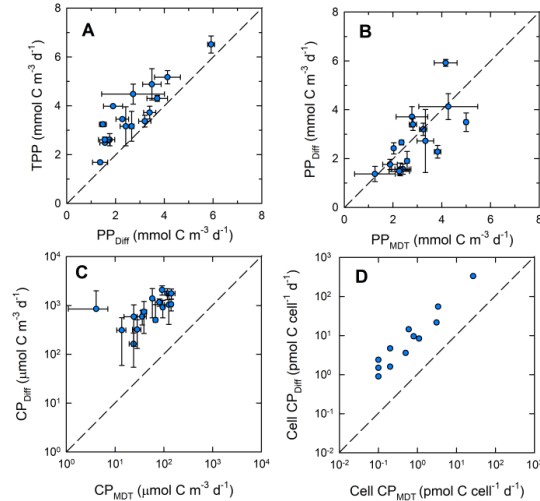

**Figure A1: Scatterplots of: (A) total particulate production (TPP) and primary production from the difference technique (PP$_{Diff}$); (B) primary production from the Micro-Diffusion Technique (PP$_{MDT}$) and difference method (PP$_{Diff}$); (C) CaCO$_3$ production from the Micro-Diffusion Technique (CP$_{MDT}$) and**

**difference method (CP$_{Diff}$); and (D) cell-normalised rates (cell-CP) from the Micro-Diffusion Technique (Cell-CP$_{MDT}$) and difference method (Cell-CP$_{Diff}$). Dashed lines in all panels indicate unity.**