# Peer review of "A global compilation of coccolithophore calcification rates"

_Earth System Science Data, 2018_

## Referee Comment (RC1) · L.T. Bach (Referee) · 28 May 2018

In this study, Daniels et al. compile all available coccolithophore calcification rates from the field in a common data base. They use published and unpublished data thereby enlarging our current data coverage substantially. The dataset is quality-controlled and invaluable for coccolithophore research, especially in light of the increasing number of studies using spaceborne and in situ sensor data to estimate calcification rate. The accompanying text is well written. I only have minor comments/suggestions.

Line 38ff: In this context it may also be useful to mention that satellites are restricted to the upper euphotic zone and miss/underestimate cocco blooms that are stretched out over the water column.

Line 54f: Why would CaCO3 export be a positive feedback on atmospheric CO2. Wouldn't the ballasting effect outweigh the Alkalinity drawdown?

Line 155: Hasn't temperature also always been measured? Temperature would be helpful.

Line 155: In Line 163 you write 6 – 24 hours. In line 216 you refer to Lam et al. who incubated 5 hours. I suggest being consistent on this in the text.

Line 167: The approximated abundances of foraminifera and pteropods would probably require a reference.

Line 246ff: Readers who are not familiar with the "optical depth" assessment/rationale (like me) may appreciate a more thorough explanation of this concept. It is a bit of a pain to go through the Kirk reference only for this.

Line 275: I totally agree with removing zero values. However, also some very low measurements which are slightly above zero may still be below the detection limit and thus unreliable. I wonder if it wouldn't be helpful to give an approximate detection limit of the method just to show how trustworthy such very low values would be.

Line 349ff: This is an interesting aspect. Can you assure that the incubations were made at light levels too low for photosynthesis (and not only the sampling depths)? Otherwise the hypothesis would not hold.

Line 359: "global CP" or "global CP by coccolithophores"? I guess it is the latter but this should be clarified here.

Line 375: I find this not particularly convincing because per cell calcification rates may vary massively in between species. So I wonder if it is useful normalize to cell abundance without accounting for the different sizes of the various species. There seem to be too many degrees of freedom to come to any particularly useful conclusions. You basically say this yourself in the subsequent paragraphs.

Line 796: 2% (10 to 28)? Do you mean 20%?

Appendix: Maybe this is just a problem for me but I found the acronym "diff" (for difference) a bit unlucky because I confused it with diffusion all the time.

Figure 4: I guess the x-axis is on a logarithmic scale? If so, it would be good to label it as such.

[Figure]

---

## Referee Comment (RC2) · Anonymous Referee #2 · 31 May 2018

Reviewer 2 comments on Daniels et al. "A global compilation of coccolithophore calcification rates"

General comments:

Daniel et al. have created a global dataset of field calcification rates by coccolithophores. This dataset will be extremely useful to the scientific community and should be published. The manuscript is well written and the figures are clear and describe the dataset well. I have a few comments and suggestions below, but overall the manuscript is in great shape and I recommend publication after minor revisions.

Specific comments:

Line 38: excessive "the" before "coccolithophores"

[Figure]

Line 63: Biocalcification is also poorly constrained due to data limitations (e.g., satellite-derived PIC only sees the surface and is tuned to capture E. hux and not other species)

Line 84-86: Might want to also mention that the E hux morphotype B/C, which dominates the Southern Ocean (Charalampopoulou et al. 2016), is particularly lightly calcified and the PIC algorithm overestimates PIC in the Southern Ocean due to the unique reflectance properties of E hux B/C (see Holligan et al., 2010)

Line 97: Unclear who "their" is referring to. Either delete it or replace it with "coccolithophore", if you are just referring to coccolithophore calcification.

Line 105: Perhaps add to this citation list: two recent reviews by Balch (Annual Review in Marine Science) and Krumhardt et al. (Progress in Oceanography) – see ref list at the end of this document for complete citation.

Line 238: There is a left open parenthesis in this sentence and it's a bit confusing. I suggest a rewrite: "From these profiles, depth-integrated values were calculated to represent euphotic zone integrated CP in which the euphotic zone is taken as either 1% (e.g. Poulton et al., 2006) or 0.1% (e.g. Balch et al., 2011) of incident irradiance in the different studies." Line 252: Is there an extra "of" after "surface"?

Line 358: I'm confused about this range of global CP estimates. It is indeed highly uncertain but 8 Gt C yr-1 seems way too high. I'm not seeing this value in the references that are cited. Another more recent reference that would be an upper end of the range would be Smith and Mackenzie, 2016 (2.1 Gt C yr-1)

Figure 5 (and maybe elsewhere): Since coccolithophores are well known to be quite seasonal, perhaps point out that the points on the maps are not separated by season but are all measurements are included on these maps regardless of the time of year the CP measurement was taken. Due to the seasonal bias in the dataset we could almost see this as a "growing season snapshot" (?)

Line 375: Is cell-CP really a measure of calcification per unit biomass? Cell size and organic carbon content in coccolithophores can vary between species and under changing environmental conditions (see POC-normalized growth rates in Krumhardt et al., 2016 and volume normalization in Muller et al., 2017)

Line 394: Southern Ocean E hux morphotype B/C approaches this low cell-CP (Figure 1i in Muller et al., 2015, converting from pgC cell-1 d-1 to pmol cell-1 d-1)

References: Balch, William M. "The Ecology, Biogeochemistry, and Optical Properties of Coccolithophores." Annual review of marine science 10, no. 1 (2018).

Holligan, P. M., A. Charalampopoulou, and R. Hutson. "Seasonal distributions of the coccolithophore, Emiliania huxleyi, and of particulate inorganic carbon in surface waters of the Scotia Sea." Journal of Marine Systems 82, no. 4 (2010): 195-205.

Krumhardt, Kristen M., Nicole S. Lovenduski, M. Debora Iglesias-Rodriguez, and Joan A. Kleypas. "Coccolithophore growth and calcification in a changing ocean." Progress in Oceanography (2017).

Müller, Marius N., Thomas W. Trull, and Gustaaf M. Hallegraeff. "Independence of nutrient limitation and carbon dioxide impacts on the Southern Ocean coccolithophore Emiliania huxleyi." The ISME journal 11, no. 8 (2017): 1777.

Müller, Marius N., Thomas W. Trull, and Gustaaf M. Hallegraeff. "Differing responses of three Southern Ocean Emiliania huxleyi ecotypes to changing seawater carbonate chemistry." Marine Ecology Progress Series 531 (2015): 81-90.

Smith, Stephen V., and Fred T. Mackenzie. "The role of CaCO3 reactions in the contemporary oceanic CO2 cycle." Aquatic geochemistry 22, no. 2 (2016): 153-175.

---

## Author Comment (AC1) · 25 Aug 2018

RC1 (L.T. Bach) In this study, Daniels et al. compile all available coccolithophore calcification rates from the field in a common data base. They use published and unpublished data thereby enlarging our current data coverage substantially. The dataset is quality-controlled and invaluable for coccolithophore research, especially in light of the increasing number of studies using space-borne and in situ sensor data to estimate calcification rate. The accompanying text is well written. I only have minor comments/suggestions.

1.1 Authors: We thank Dr Bach for his positive comments and clear statement on the value of the dataset. We respond to his comments below, noting that many relate to

our wish to keep the ESSD manuscript short and simple.

Line 38ff: In this context it may also be useful to mention that satellites are restricted to the upper euphotic zone and miss/underestimate cocco blooms that are stretched out over the water column.

1.2 Authors: We have revised the abstract accordingly: "Satellite-based estimates of areal CP are limited to surface waters and open-ocean areas, with current algorithms utilising the unique optical properties of the cosmopolitan bloom-forming species Emiliania huxleyi, whereas little understanding of deep-water dynamics or the optical properties of..".

Line 54f: Why would $CaCO_3$ export be a positive feedback on atmospheric $CO_2$. Wouldn't the ballasting effect outweigh the Alkalinity drawdown?

1.3 Authors: While we agree with the reviewer that ballasting could potentially alter the magnitude, or even the sign, of the overall biogeochemical impact of coccolithophores on atmospheric $CO_2$. We also note that global ballasting of POC flux by $CaCO_3$ is hypothesised rather than a fully resolved mechanism (e.g. sticky organic matter may drag down minerals rather than vice versa), and there are also timescale issues to consider (i.e. seasonal versus geological), whereas the chemical effect of calcification is straightforward chemistry. In addition, over the longer term POC export is likely to happen eventually either way, whether coccolithophores (and $CaCO_3$) are present or not, given that phytoplankton blooms tend to continue until nutrients are exhausted. Hence, on a global scale the total amount of POM exported over a bloom season is more a function of nutrient supply than mineral availability.

We accept however that there are uncertainties about the overall biogeochemical impact on atmospheric $CO_2$, but have left the sentence unaltered since our sentence refers only to the impact of calcification plus export and not to the overall impact.

Line 155: Hasn't temperature also always been measured? Temperature would be

helpful.

1.4 Authors: Temperature (and nutrient concentrations) would indeed be very helpful, however it was not always reported (or made available) for many of the datasets collated from the literature or unpublished sources. Furthermore, ensuring that we can provide quality controlled ancillary data is also challenging, whereas internally we could ensure that the rate data was adequately quality controlled.

Line 155: In Line 163 you write 6 – 24 hours. In line 216 you refer to Lam et al. who incubated 5 hours. I suggest being consistent on this in the text.

1.5 Authors: Corrected to 5 to 24 h now in section 2.1.2.

Line 167: The approximated abundances of foraminifera and pteropods would probably require a reference.

1.6 Authors: We have now revised and added references to this line as follows:

'..and rare calcifying organisms, such as foraminifera (typically $\leq$0.5 L-1; e.g. Schiebel & Movellan, 2012) or pteropods (typically $\leq$0.005 L-1; Burridge et al., 2017)

References added:

Burrdige, A.K., Goetze, E., Wall-Palmer, D., Le Double, S.L., Huisman, J., Peignenburg, K.T.C.A.: Diversity and abundance of pteropods and heteropods along a latitudinal gradients across the Atlantic Ocean, Prog. Oceanography, 158, 213-223, 2017.

Schiebel, R., and Movellan, A.: Global distributions of modern planktic foraminifera abundance and biomass – Gridded data product (NetCDF) – Contribution to MAREDAT World Ocean Atlas of Plankton Functional Types, PANGAEA, https:/doi.org/10.1594/PANGAEA.777386, 2012.

Line 246ff: Readers who are not familiar with the "optical depth" assessment/rationale (like me) may appreciate a more thorough explanation of this concept. It is a bit of a pain to go through the Kirk reference only for this.

1.7 Authors: We have now added the following explanation (@ Ln 246): 'Optical depth represents the path length of light through a medium and is the natural logarithm of the ratio of surface irradiance to irradiance at a specific depth, being proportional to the amount of light attenuation in the water column. Consideration of optical depth rather than absolute depth accounts for geographical patterns in the light field, recognising light as an important driver of CP. For example, 1% of surface irradiance (optical depth of 4.6 as natural log of 0.01) may reach 30 m in temperate waters with high attenuation, whereas it may reach 90 m in subtropical waters with low attenuation: if incidental irradiance was the same at both sites then both depths would receive the same light intensity independent of the difference in depth.'

Line 275: I totally agree with removing zero values. However, also some very low measurements which are slightly above zero may still be below the detection limit and thus unreliable. I wonder if it wouldn't be helpful to give an approximate detection limit of the method just to show how trustworthy such very low values would be.

1.8 Authors: The MDT method includes a formalin-killed blank whereby each triplicate light incubated rate measurement has a blank value subtracted, with the blank accounting for any abiotic isotope uptake. Hence, zero values represent CP rates where the live values are equal (or less) than the abiotic values and any value above zero has a positive measurable calcification rate. As the methodology of the measurements has no standard in terms of radioactive spike, volume incubated or incubation length, the relationship between live values and formalin-blank values can be very variable between studies. Hence, it is not possible to approximate a standard detection limit as it will vary between studies depending on aspects of their methodology; it is possible to lower the detection limit (and increase sensitivity) to some extent through increasing the magnitude of the radioactive spike, though we also note that in some cases this can also lead to elevated blank values.

Line 349ff: This is an interesting aspect. Can you assure that the incubations were made at light levels too low for photosynthesis (and not only the sampling depths)?

[Figure]

Otherwise the hypothesis would not hold.

1.9 Authors: The incubation irradiances are not contained in the database. However, Poulton et al. (2017) estimate that irradiance levels below the subtropical DCM (at 1% of surface irradiance) are less than 20 $\mu$mol photons m-2 s-1 (equivalent to <1.2 mol quanta m-2 d-1, assuming a 16 h day), an irradiance level often seen to light-limit cultures and North Atlantic phytoplankton (e.g. Siegel et al., 2002). We have now slightly reworded this sentence: 'CP at depths below the light levels for photosynthetic growth may...'.

Siegel, D.A., Doney, S.C., Yoder, J.A.: The North Atlantic spring phytoplankton bloom and Sverdrup's critical depth hypothesis. Science, 296, 730-733, 2002.

Line 359: "global CP" or "global CP by coccolithophores"? I guess it is the latter but this should be clarified here.

1.10 Authors: Indeed, now corrected to 'global CP by coccolithophores'.

Line 375: I find this not particularly convincing because per cell calcification rates may vary massively in between species. So I wonder if it is useful normalize to cell abundance without accounting for the different sizes of the various species. There seem to be too many degrees of freedom to come to any particularly useful conclusions. You basically say this yourself in the subsequent paragraphs.

1.11 Authors: Indeed, the influence of different cell calcite content on cell-normalised calcification can be significant, and this is in effect what we are suggesting – normalising calcification rates to cell numbers give you an idea of the community-wide calcification rate per cell numbers; in a E. huxleyi community, cell-normalised calcification rate can infer physiological changes and controlling factors, in a diverse community it infers the compositional influence on calcification rates. A large fraction of the CP measurements have been collected from waters that can be considered E. huxleyi dominated, and so in these cases there is the possibility to examine cell-CP in light of variability in

environmental conditions (see e.g. Poulton et al., 2010, 2013; Charalampopoulou et al., 2016).

Ideally it would be good to normalise measurements of CP to community calcite (from cells, not detrital) and examine differences in growth rates (see Poulton et al., 2010); however, this has several difficulties for mixed communities in terms of weighted means and potential variability in growth rates and relative abundances (see Daniels et al., 2016). In light of these (potentially current) difficulties we have normalised to cell abundance as a first order measure of (a) whether the CP rates are physiologically realistic, and (b) a (admittedly very rough) index of the relative CP rates between different communities.

Our central tenant in terms of normalising to cell numbers is to provide a first-order check on whether the CP rates are physiologically possible, and encourage future studies to look in more depth into the coccolithophore community responsible for these rates.

Line 796: 2% (10 to 28)? Do you mean 20%?

1.12 Authors: We are not sure where this error crept in but this sentence should read: PPMDT measurements was 14% (2 to 66%) and 19% (1 to 72%) for CPMDT, across a range of PPMDT from 1.3 to 5.0 mmol C m-3 d-1.'

Appendix: Maybe this is just a problem for me but I found the acronym "diff" (for difference) a bit unlucky because I confused it with diffusion all the time.

1.13 Authors: We have now changed to FF for the difference (diFF).

Figure 4: I guess the x-axis is on a logarithmic scale? If so, it would be good to label it as such.

1.14 Authors: Figure 4 and Figure 5B have now been changed to reflect this.
* * *
[Figure]

2018.

---

## Author Comment (AC2) · 25 Aug 2018

RC2 (Anonymous)

General comments: Daniel et al. have created a global dataset of field calcification rates by coccolithophores. This dataset will be extremely useful to the scientific community and should be published. The manuscript is well written and the figures are clear and describe the dataset well. I have a few comments and suggestions below, but overall the manuscript is in great shape and I recommend publication after minor revisions.

2.1 Authors: We thank the reviewer for their positive comments and respond to their specific comments below.

[Figure]

Specific comments:

Line 38: excessive "the" before "coccolithophores"

2.2 Authors: Changed.

Line 63: Biocalcification is also poorly constrained due to data limitations (e.g., satellite derived PIC only sees the surface and is tuned to capture E. hux and not other species)

2.3 Authors: This is already covered later in this section (Lns 75-94).

Line 84-86: Might want to also mention that the E hux morphotype B/C, which dominates the Southern Ocean (Charalampopoulou et al. 2016), is particularly lightly calcified and the PIC algorithm overestimates PIC in the Southern Ocean due to the unique reflectance properties of E hux B/C (see Holligan et al., 2010)

2.4 Authors: Now added to Lns 88 to 89.

Line now reads:

'Relatively small differences in the CaCO3 content of the various E. huxleyi morphotypes (Young et al., 2003; Poulton et al., 2011; Charalampopoulou et al., 2016) can have significant impacts in terms of the satellite-retrieval of PIC concentrations (Holligan et al., 2010) and CaCO3 formation at the scale of mesoscale blooms (Poulton et al., 2013).

Reference added:

Holligan, P.M., Charalampopoulou, A., and Hutson, R: Seasonal distribution of the coccolithophore Emiliania huxleyi and of particulate inorganic carbon in the surface waters of the Scotia Sea, J. Mar. Sys., 82, 195-205, 2010.

Line 97: Unclear who "their" is referring to. Either delete it or replace it with coccolithophore", if you are just referring to coccolithophore calcification.

2.5 Authors: Apologies, a mistake on our part. 'Their' is now deleted.

Line 105: Perhaps add to this citation list: two recent reviews by Balch (Annual Review in Marine Science) and Krumhardt et al. (Progress in Oceanography) – see ref list at the end of this document for complete citation.

2.6 Authors: Both now added to this section.

Ln now reads:

'. . .reviewed numerous times (see Paasche, 2002; Zondervan, 2007; Boyd et al., 2010; Raven and Crawfurd, 2012; Monteiro et al., 2016, Taylor et al., 2016' Krumhardt et al., 2017; Balch et al., 2018).'

References added:

Balch, W.M.: The ecology, biogeochemistry and optical properties of coccolithophores, Ann. Rev. Mar. Sci. 10, 71-78, 2018.

Krumhardt, K.M., Lovenduski, N.S., Iglesias-Rodriguez, M.D., and Kleypas, J.A.: Coccolithophore growth and calcification in a changing ocean, Prog. Oceanography, 159, 276-295, 2017

Line 238: There is a left open parenthesis in this sentence and it's a bit confusing. I suggest a rewrite: "From these profiles, depth-integrated values were calculated to represent euphotic zone integrated CP in which the euphotic zone is taken as either 1% (e.g. Poulton et al., 2006) or 0.1% (e.g. Balch et al., 2011) of incident irradiance in the different studies."

2.7 Authors: We agree and have changed as suggested.

Line 252: Is there an extra "of" after "surface"?

2.8 Authors: Yes, now removed.

Line 358: I'm confused about this range of global CP estimates. It is indeed highly uncertain but 8 Gt C yr-1 seems way too high. I'm not seeing this value in the references

that are cited. Another more recent reference that would be an upper end of the range would be Smith and Mackenzie, 2016 (2.1 Gt C yr-1)

2.9 Authors: We apologise for a typo in this line, 8 Gt C yr-1 should read 1.8 Gt C yr-1 (based on Berelson et al. (2007)'s range of modelled PIC export from the surface). Overall this section aims to highlight the uncertainty in current budgets and provide a simple estimate of CP from the in situ measurements (rather than provide a full overview of global CP). The 2.1 Gt C yr-1 by Smith & Mackenzie (2016) actually includes both plankton production (1.6 Gt C yr-1) and shelf benthos production (0.5 Gt C yr-1). To avoid confusion, we now give the range as 0.4 to 1.6 Gt C yr-1 (to reflect the general agreement between Balch et al. (2007), Berelson et al. (2007) and Smith & Mackenzie (2016)) for pelagic plankton production.

The line now reads:

'.. with estimates ranging from 0.4 to 1.6 Gt C yr-1 (Balch et al., 2007; Berelson et al., 2007; Smith & Mackenzie, 2016).'

Reference added:

Smith, V.S., and F.T. Mackenzie: The role of CaCO3 reactions in the contemporary oceanic CO2 cycle. Aquatic Geochemistry, 22, 153-175, 2016.

Figure 5 (and maybe elsewhere): Since coccolithophores are well known to be quite seasonal, perhaps point out that the points on the maps are not separated by season but are all measurements are included on these maps regardless of the time of year the CP measurement was taken. Due to the seasonal bias in the dataset we could almost see this as a "growing season snapshot" (?)

2.10 Authors: Indeed, with this in mind we have now added a line to the Figure 5 legend (see below) and a new line (see below) in section 3.2.1 (@ Ln 307). In terms of 'growing season snapshot', it would be tempting to agree and overplay this but the 25 year period over which the measurements have been collected warrants a very careful

consideration of such a 'snapshot'.

Line added (@ Ln 307):

'Some of this variability arises due to a lack of temporal resolution in Figure 5, where 25 years of measurements are plotted alongside one another, with a recognisable seasonal bias towards summer in both hemispheres (Section 3.1).'

Figure 5 Legend now reads:

'Global maps of (A) surface CaCO3 production (CP), and (B) euphotic zone integrated CP. Global maps superimposed on ocean bathymetry as in Fig. 1A. Note: Global maps represent all measurements and are not temporally resolved.'

Line 375: Is cell-CP really a measure of calcification per unit biomass? Cell size and organic carbon content in coccolithophores can vary between species and under changing environmental conditions (see POC-normalized growth rates in Krumhardt et al., 2016 [2017] and volume normalization in Muller et al., 2017)

2.11 Authors: Indeed, many measures of coccolithophore cell biomass (cell size, biovolume, organic content, inorganic content) do vary with environmental conditions, as well as having a natural population range (rather than a fixed value) for all species in mixed communities. None of these are routinely measured (or reported), whereas most often coccolithophore cell counts do accompany coccolithophore studies. There are also recognised difficulties in estimating cell (organic) biomass (or coccosphere biovolume) for diverse field communities (O'Brien et al., 2016) or in the case of culture-based studies from extant bacteria.

Ideally it would be good to normalise measurements of CP to community calcite (from cells, not detrital) and examine differences in growth rates (see Poulton et al., 2010); however, this has several difficulties for mixed communities in terms of weighted means and potential variability in growth rates and relative abundances (see Daniels et al., 2016). In light of these (potentially current) difficulties we have normalised to cell abun-

dance as a first order measure of (a) whether the CP rates are physiologically realistic, and (b) a (admittedly very rough) index of the relative CP rates between different communities.

Line 394: Southern Ocean E hux morphotype B/C approaches this low cell-CP (Figure 1i in Muller et al., 2015, converting from pg C cell-1 d-1 to pmol cell-1 d-1)

2.12 Authors: Good point, though we note that in this case the growth rates are relatively low (<0.1 d-1).

We have now changed Ln 394 to: 'Thus, samples in Figure 8 with a cell-CP lower than this value could be dominated by slow growing (<0.1 d-1) low-calcite morphotypes of E. huxleyi (see Müller et al., 2015) or coccolithophore species with much lower . . .'.

Reference added:

Müller, M.N., Trull, T.W., and Hallegraeff, G.M.: Differing responses of three Southern Ocean Emiliania huxleyi ecotypes to changing seawater carbonate chemistry, Mar. Ecol. Prog. Ser. 531, 81-90, 2015.